# Evolutionary genomics of camouflage innovation in the orchid mantis

Guangping Huang [1,4], Lingyun Song[1,2,4], Xin Du[1,2], Xin Huang[1,2] & Fuwen Wei [1,2,3] ✉

The orchid mantises achieve camouflage with morphological modifications in body color and pattern, providing an interesting model for understanding phenotypic innovation. However, a reference genome is lacking for the order Mantodea. To unveil the mechanisms of plant-mimicking body coloration and patterns, we performed de novo assembly of two chromosome-level genomes of the orchid mantis and its close relative, the dead leaf mantis. Comparative genomic analysis revealed that the *Scarlet* gene plays an important role in the synthesis of xanthommatin, an important pigment for mantis camouflage coloration. Combining developmental transcriptomic analysis and genetic engineering experiments, we found that the cuticle was an essential component of the 'petal-like' enlargement, and specific expression in the ventral femur was controlled by Wnt signaling. The prolonged expression of Ultrabithorax (Ubx) accompanied by femoral expansion suggested that *Ubx* determines leg remodeling in the early developmental stage. We also found evidence of evolution of the *Trypsin* gene family for insectivory adaptation and ecdysone-dependent sexual dimorphism in body size. Overall, our study presents new genome catalogs and reveals the genetic and evolutionary mechanisms underlying the unique camouflage of the praying mantis, providing evolutionary developmental insights into phenotypic innovation and adaptation.

As predator–prey interactions are a substantial component of all biological communities, camouflage arising from natural selection is a key adaptive strategy for avoiding predators or attracting prey[1–4]. Animal camouflage involves morphological modifications in body colors and patterns that reduce visual detection or recognition[5,6]. To date, most studies have focused on antipredator camouflage, as a wide range of prey animals benefit from masquerading as inedible and often inanimate objects, such as twigs, leaves, stones or bird droppings[7]. However, very little attention has been given to predatory camouflage strategies[8].

Many clades of predatory insects exhibit a remarkable range of morphological camouflage adaptations in response to predation pressure and to attract prey[9–11]. One of the most striking, yet poorly understood, cases occurs in the Mantodea, which are carnivorous insects that exhibit rich diversity, with approximately 2500 species possessing diverse morphological and ecological characteristics[12]. Despite their popularity and appeal, praying mantises are understudied. Many species in this lineage use strategies involving spectacular morphological modifications in body color and pattern to mimic plant parts, such as flowers, dead leaves, sticks, and barks, and mosses[13].

The orchid mantis *Hymenopus coronatus* (Mantodea: Hymenopodidae), a popular and charismatic species, has developed spectacular flower-mimicking strategies[12,14,15]. The white and pink coloration in combination with its "petal-shaped" femoral lobes and broad

[1]CAS Key Laboratory of Animal Ecology and Conservation Biology, Institute of Zoology, Chinese Academy of Sciences, Beijing 100101, China. [2]University of Chinese Academy of Sciences, Beijing 100049, China. [3]College of Forestry, Jiangxi Agricultural University, Nanchang 330045, China. [4]These authors contributed equally: Guangping Huang, Lingyun Song. ✉e-mail: weifw@ioz.ac.cn

abdomen allow the orchid mantis to mimic generalized flowers. Interestingly, this floral simulation alone attracts more insect pollinators than real flowers[16]. The trade-off between antipredator camouflage and predatory camouflage of the orchid mantis makes it a master of camouflage. Another popular plant-mimicking mantis is the dead leaf mantis *Deroplatys lobata* (Mantodea: Deroplatyidae), which shows coloring ranging from brown to gray and possesses a broad prothorax that looks like a ripped and crumpled leaf[17]. The false leaf vein on the adult's tucked wings and leaf petiole residue-like legs further its dead-leaf-like appearance. In addition to the uncanny combination of body colors and patterns, these mantises also display features generalizable to any mantis, including insectivory and extreme sexual dimorphism in body size[18,19]. However, the genetic mechanisms underlying these unique phenotypes remain unexplored. The lack of genomic resources for Mantodea lineages has impeded further in-depth molecular and developmental investigation.

In this work, we preform de novo-assembly of the chromosome-level genomes of the flower-mimicking species *H. coronatus* with dead leaf-mimicking species *D. lobata* to investigate and compare the genetics and evolutionary mechanism of phenotypic innovation of these species. Comparative genomic and developmental transcriptomic analyses, and genetic engineering experiments based on RNA interference (RNAi) were performed to dissect the genetic basis of these camouflage phenomena. Here, we present reference genomic resources for two species of mantis and explore the genetic mechanisms of adaptive camouflage in the orchid mantis via the combination of genomic and transcriptomic analysis with functional testing. We believe these results will have far-reaching implications for future in-depth study of Mantodea.

## Results and Discussion
### Genomic signatures in two highly camouflaged mantises

Due to the highly camouflaged appearances of the orchid mantis and dead leaf mantis, they are rarely observed in the wild (Fig. 1A), resulting in little knowledge available for their evolution and biology. To decode the genomic signature of two plant-mimicking mantises, *H. coronatus* and *D. lobata*, we performed single-molecule real-time sequencing (Nanopore), Illumina paired-end sequencing, and Hi-C sequencing (Fig. 1B). The high-quality Illumina reads were used to correct the short insertions or deletions (indels) and substitutions in the Nanopore assemblies. More than 98.13% of the Hi-C reads were successfully anchored to 23 chromosomes for *H. coronatus* (2n = 46) and 14 chromosomes for *D. lobata* (2n = 28) (Supplementary Fig. 1; Supplementary Table S1). Finally, two genome assemblies were obtained with sizes of 2.88 Gb for *H. coronatus* and 3.96 Gb for *D. lobata*, showing contig N50 lengths of 15.76 Mb and 6.11 Mb, respectively (Supplementary Table 2). Subsequently, the integrity of the assembly was demonstrated by 99.32%–99.87% mapping rates for Illumina sequencing reads and Nanopore sequencing reads (Supplementary Table 3) and 96.56%–96.74% Benchmarking Universal Single-Copy Orthologs (BUSCO) completeness (Supplementary Table 4). A series of evaluations indicated that the genome assemblies were of high quality (Supplementary Fig. 2).

The genomes of the mantises were larger than most of those sequenced for heterometabolous insects, such as *B. germanica* (1.79 Gb) and *Z. nevadensis* (0.48 Gb), containing a high level of repetitive elements amounting to 64.94–66.16% of the genome assemblies (Supplementary Table 5). Among them, 42.36% and 51.92% of the genome sequences were transposable elements (TEs) (1.21 Gb for *H. coronatus* and 2.06 Gb for *D. lobata*). The class II retrotransposon DNA transposons (558.37 Mb, 19.39% of genomes) represented the most dominant TE elements, with TcMar-Tc1 being the largest subclass (Fig. 1C, Supplementary Table 5). In addition, class I retrotransposons, including long interspersed nuclear elements (LINEs), short interspersed nuclear elements (SINEs), and long terminal repeats (LTRs),

constituted 20.80%–22.39% of the genomes (Supplementary Table 5). Linear regression analysis showed that the length of DNA transposons was positively correlated with the variation in genome size (Supplementary Fig. 3), consistent with the findings of a previous study[20], indicating the important role of retrotransposons in mantis genome expansion.

Based on the transcript dataset combining ab initio and homology-based approaches, we identified 16,294 and 17,691 protein-coding genes with average lengths of 77.17–83.96 kb in these two mantis genomes, containing 95.5%–98.2% of complete conserved orthologs within insects (Supplementary Tables 6 and 7). More strikingly, the average length of introns was 12.77–14.97 kb, larger than that in other insect taxa, while the average exon length (218.37–222.83 bp) was similar to that in other insects, resulting in the extension of gene length (Fig. 1D, Supplementary Fig. 4). Finally, ~90% of the genes were functionally annotated (Supplementary Table 8). Analysis of gene family evolution revealed 794 and 1045 expanded gene families in the orchid and dead leaf mantises, respectively (Fig. 1E).

Taken together, these results present two high-quality chromosome-level genome assemblies and gene catalogs for the Mantodea, enabling further insights into the biology and evolution of these extraordinary camouflaged mantises.

### Resolving the Mantodea phylogenetic position and evolutionary history

Phylogenomic analysis was performed to test the phylogenetic position of Mantodea. 324 single-copy orthologs identified in these two mantises and 16 other insects from 15 key extant insect orders were used to reconstruct a phylogenetic tree (Fig. 1E, Supplementary Fig. 5). The phylogenetic result was congruent with a recent insect phylogeny[21], with all the bootstrap values of the concatenated tree above 96% (Fig. 1E). The phylogenetic analysis supported the most recent phylogenetic hypothesis that Mantodea is a sister group to Blattodea, and their divergence was estimated to approximately the Permian–Triassic period (211.89–277.30 million years ago (Ma)) (Fig. 1E), in agreement with the divergence date estimated by morphological and molecular data[22].

The time of divergence of flower- and dead leaf-mimicking camouflage is estimated to be 36.50 Ma ago [95% highest posterior density (HPD) 24.15–55.31 Ma] (Fig. 1E), following the transition to widespread angiosperm-dominated biomes in the Paleocene (66–56 Ma)[23]. Insect pollination, as the dominant mode of angiosperm pollination, probably drove the diversification and ecological expansion of flower-mimicking mantises[24].

To better understand the demographic history of plant-mimicking camouflage in Mantodea, pairwise sequential Markovian coalescent (PSMC) analysis was performed to assess the evolutionary history and dynamic effective population size[25]. *H. coronatus* and *D. lobata* populations underwent stepwise expansion in the early Pleistocene (Supplementary Fig. 6). Subsequently, *H. coronatus* experienced a population reduction at approximately 1 Ma, whereas the *D. lobata* population experienced an expansion approximately 300 thousand years ago (kya) and subsequently declined, coinciding with dramatic temperature changes, such as intensified amplitudes of glacial cycles during the Penultimate Glaciation (PG, 200–130 kya) and the Last Glaciation (LG, 70–10 kya) (Supplementary Fig. 6). The distribution of the orchid mantis has been reported in the rainforest across Malaysia, Indonesia, India, Thailand, Vietnam and Southern China[15], while that of the dead leaf mantis is mainly in the rainforest of Malaysia and Philippines[26]. Both of them showed intrinsic susceptibility to extreme environmental fluctuations. The distinct demographic histories between the two mantis species demonstrated that differential coloration probably confers host-heterogeneous responses to environmental change; a similar finding has been reported in the oakleaf butterfly[27]. Overall, these results provide a reliable

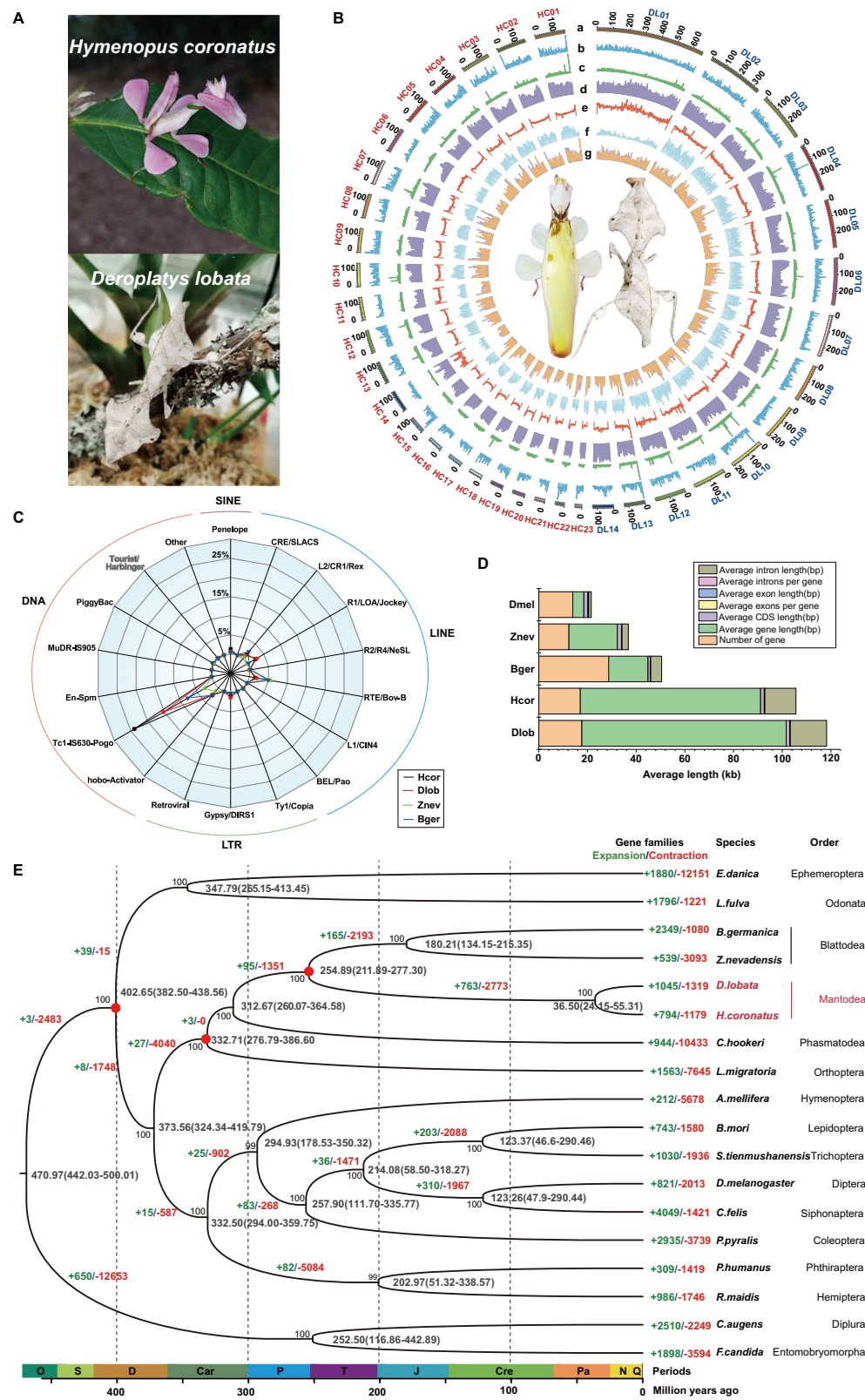

phylogenetic framework and date estimates for a better understanding the origin and evolution of praying mantis camouflage.

## Genetic and biochemical bases of camouflage coloration

Animal colors have been an intriguing scientific issue for centuries[28]. The orchid mantis displays unique color variation over the course of its development (Fig. 2A). The first-instar larva (L1) depends largely on its aposematic coloration to evade predators, mimicking the larva of a poisonous red and black assassin bug (Hemiptera: Reduviidae), exhibits red and black aposematic colors and subsequently changes to a white and pink color pattern starting in the second-instar larval stage (L2), beginning to mimic a flower blossom (Fig. 2A). This floral resemblance is most apparent in juvenile orchid mantises that have yet to develop wings. The adult orchid mantis resembles withering

**Fig. 1 | Genomic structure and phylogeny of praying mantises. A** Photos of the juvenile orchid mantis *H. coronatus* (upper) and adult dead leaf mantis *D. lobata* (down) showing their adaptive camouflage. **B** Circos plot of chromosome-level genome assemblies of *H. coronatus* and *D. lobata*. From the inner circle to the outer circle, genome characteristics are indicated successively, including chromosome sizes (a), gene density (b), noncoding RNA density (c), repeat sequence distribution (d), GC content (e), SNP density (f), and homeobox gene clusters (g).
**C** Composition comparison of the repetitive sequences of two mantises and the close relatives *Blattella germanica* and *Zootermopsis nevadensis*. The coordinates represent the percentage of the length of each category relative to the corresponding genome size. **D** Gene structure analysis showing that *H. coronatus* and *D.*

*lobata* have larger average intron lengths than *B. germanica*, *Z. nevadensis* and *Drosophila melanogaster*. Abbreviations used throughout: Hcor, *H. coronatus*. Dlob, *D. lobata*. Bger, *B. germanica*. Znev, *Z. nevadensis*. Dmel, *D. melanogaster*.
**E** Phylogenomic position of Mantodea. The estimated divergence times and the numbers of families with gene expansions (green) and contractions (red) are displayed on the branches of the phylogenetic tree. The 95% confidence interval of divergence time is indicated in brackets. Three calibration points (red circles) were applied as normal priors. The bootstrap values of most nodes were 100%. O, Ordovician. S, Silurian. D, Devonian. Car, Carboniferous. P, Permian. T, Triassic. J, Jurassic. Cre, Cretaceous. Pa, Paleogene. N, Neogene. Q, Quaternary.

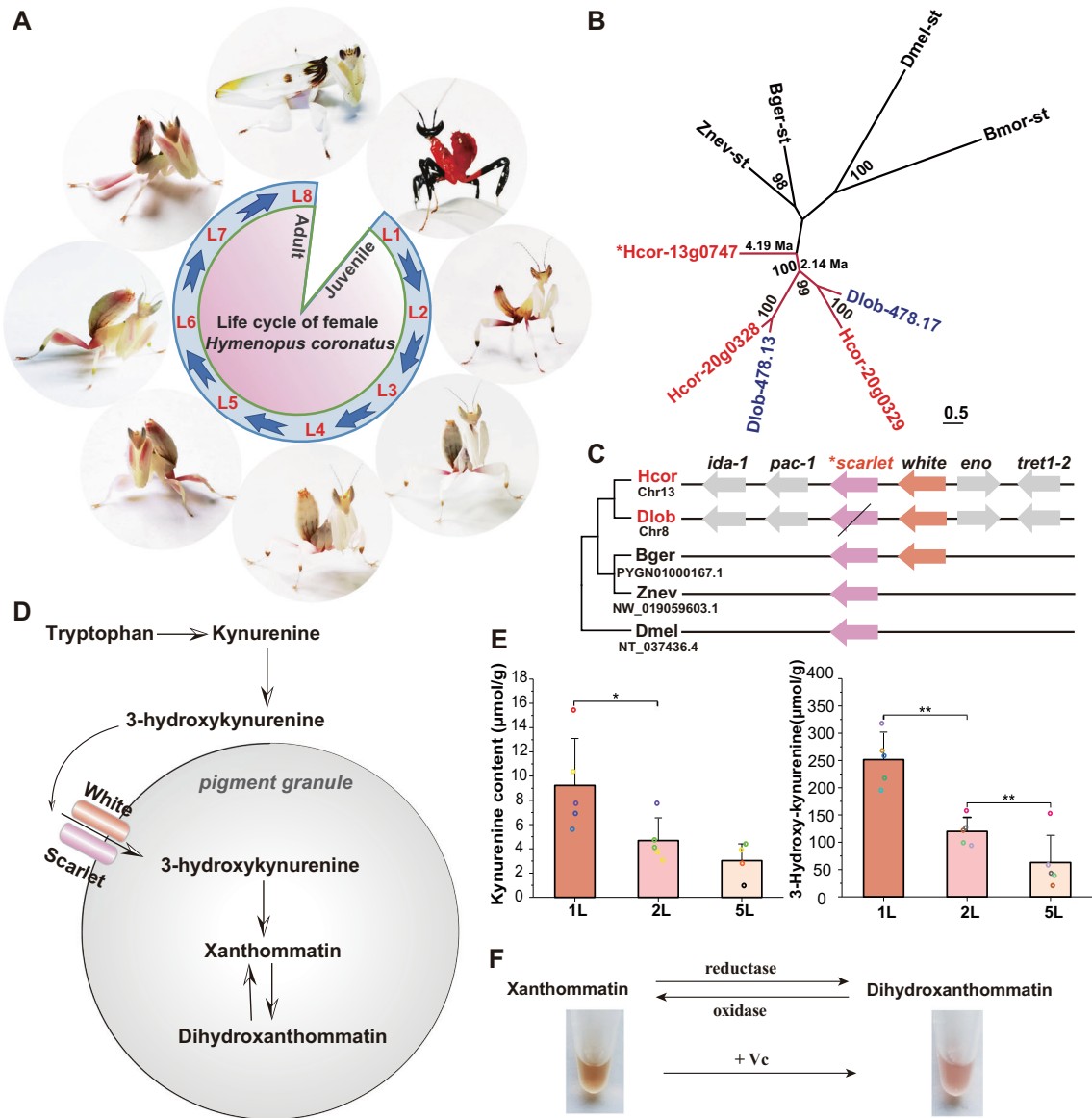

**Fig. 2 | Genetic and biochemical bases of camouflage coloration. A** Body color variation during the lifespan of the female orchid mantis shows aposematism in the first instar and camouflage coloration from the onset of the second instar.
**B** Phylogenetic analysis of the *Scarlet* gene. The degree of support for each node is shown as a probability percentage. The estimated divergence times of the *Scarlet* gene are displayed on the branches. **C** The *Scarlet* (Hcor-13g0747, marked with an asterisk to represent an orthologous gene among Insecta) and *White* gene loci.
**D** Schematic of the insect xanthommatin synthesis pathway in pigment granules.

**E** The concentrations of the substrates for xanthommatin synthesis, kynurenine and 3-hydroxy-kynurenine, were determined using high-performance liquid chromatography across developmental stages. Abbreviations used throughout: L1, first-instar larva. L2, second-instar larva. L5, fifth-instar larva. Data represent the mean ± standard deviation of five replicates. *$p = 0.04$, **$p = 0.0006$ and 0.0009, respectively, by two-sided Student's *t* tests. (**F**) In vitro redox reactions with the pigment extracted from the L2 orchid mantis. Vc, Ascorbic acid.

flowers. These results demonstrate that the orchid mantis changes its body color throughout its lifespan to successively mimic poisonous reduviids, blossoming flowers and withering flowers.

Histological analysis showed a black color in the exoskeleton, which consisted of three layers (epicuticle, exocuticle, and endocuticle), while a red color was mainly distributed in the epidermis under the exoskeleton (Supplementary Fig. 7A, B). This finding was also supported by the black pigment being retained in the cuticle shell of L1 after molting (Supplementary Fig. 7C, D). In addition, transmission electron microscopy observations confirmed the presence of pigments in the endocuticle of the red-body color area (Supplementary Fig. 8).

Gene family analysis showed that the ABCG gene family was significantly expanded in *H. coronatus* (Supplementary Fig. 9). The ABCG subfamily belongs to the ABC superfamily and plays an important role in compound transport[29]. Within the ABCG subfamily, the *Scarlet* gene underwent an expansion in both sequenced mantises, with three and two copies in *H. coronatus* and *D. lobata*, respectively. Two duplications of *Scarlet* occurred at approximately 4.19 Ma and 2.14 Ma (Fig. 2B). Comparative genome analysis showed highly conserved synteny in *Scarlet* and *White* genes among *B. germanica*, *H. coronatus* and *D. lobata* (Fig. 2C). Notably, eight exons of the *Scarlet* gene were lost in *D. lobata* (Supplementary Fig. 10), which might explain the deficiency of a red or pink color in *D. lobata*. The Scarlet protein in combination with the White protein, forms a heterodimer responsible for transporting 3-hydroxykynurenine into pigment granules to synthesize xanthommatin (Fig. 2D), which is a widespread invertebrate pigment belonging to the ommochromes[30].

To further validate the role of xanthommatin in the formation of red and pink colors, the concentrations of the substrates 3-hydroxykynurenine and kynurenine were determined in L1, L2, and fifth-instar (L5) larvae by high-performance liquid chromatography (Supplementary Fig. 11). A gradient decrease in the concentration was observed along with fading of the red body color during instar development (Fig. 2E). Xanthommatin pigments can change color under oxidative/reductive conditions: decarboxylated xanthommatin is yellow under oxidative conditions, and xanthommatin is pale red or crimson red under reductive conditions[31]. To validate the important role of xanthommatin in the formation of the pink color pattern, we further extracted pigment from L2 *H. coronatus* individuals and treated it in vitro with redox agents. Similar redox-dependent color changes were observed as the color changed from red to pink with the addition of ascorbic acid (Fig. 2F). These results suggest that xanthommatin is an important pigment for camouflage coloration in mantises that can display a redox-dependent red or pink color change. The ABCG subfamily plays an important role in the synthesis of xanthommatin in *H. coronatus* and *D. lobata*.

## Molecular mechanism underlying the morphogenesis of plant-mimicking femoral appendages

The large, flat expansions of the exoskeleton (petal-like femoral lobes) on the femur of the mid and hind legs contribute to the resemblance of the orchid mantis to flowers (Fig. 3A), while the closely related dead leaf mantis also displays leaf petiole residue-like appendages (Fig. 1A). To investigate the makeup of these traits, we performed histological analysis and revealed that the petal-like structures were mainly composed of exoskeleton, along with connective tissues (Supplementary Fig. 12). The cuticle is the major component of insect exoskeletons, and the encoding genes underwent significant expansion in *H. coronatus* and *D. lobata*, especially in the former (Fig. 3B). Whole-genome duplication analysis revealed that *Cuticle* genes were tandemly duplicated, with 35 copies on chromosome 3 (Supplementary Fig. 13). Compared to those in *H. coronatus* L1, the femoral lobes in the mid and hind legs in L2 were enlarged to resemble petals (Fig. 2A). Integrating evolutionary biology and developmental biology are crucial for

bettering understanding the molecular basis underlying key phenotypic innovations[32]. We performed transcriptome sequencing across the developmental stages of *H. coronatus*, including L1, L2, and L5 larval stages as well as the adult stage. Comparative analysis revealed that L2 and L5 possessed significantly higher expression of *Cuticle* in the leg tissues than L1 (Fig. 3C).

We also found a higher expression level of the homeodomain transcription factor *Ultrabithorax* (*Ubx*) in L1 and L2 leg tissues than in L5 leg tissues (Fig. 3C). *Ubx* has been proven to specify segment identity along the animal body axis and controls the expression of genes that regulate tissue growth and patterning during early development. We also compared the size of mesothoracic (T2) and metathoracic (T3), and detected the expression level of *Ubx* in femur, tibia and tarsus, respectively. We found that the T3 tibia and tarsus of *H. coronatus* were enlarged both in length and area with higher expression level of *Ubx* mRNA compared to their T2 counterparts (Supplementary Fig. 14A, B), consistent with the trend observed in *Tenodera aridifolia*[33]. We detected a higher expression level of *Ubx* in T3 femur than that in T2, consistent with the observation of more enlarged T3 femur (Supplementary Fig. 14C). Compared to *T. aridifolia*, *H. coronatus* exhibits "petal-like" femoral expansion from onset of L2 stage. It was remarkably that we found that a high mRNA expression lasted in the L2 stage in T2 and T3, which might result in the occurrence of "petal-like" cuticular enlargement (Supplementary Fig. 14D). These data suggest that the prolonged expression of Ubx in the femur in the early developmental stage plays a key role in the modulation of the petal-like femoral enlargement.

We also observed significant upregulation of the extracellular matrix (ECM)-related gene *Mmp2*, the ECM protease-encoding gene *Stubble*, and the downstream RhoA signaling-related genes in the leg tissue of L2 and L5 (Fig. 3C), which are known to be important for epithelial morphogenesis of leg imaginal discs[34].

To further investigate why flat expansion occurred along the ventral axis of the femur, we then focused on the Wnt signaling pathway, which is known to play important roles in axis elongation during leg development[35]. Correspondingly, we found that genes related to the Wnt signaling pathway were significantly differentially expressed between L1 and L5. Compared to those in L1, Wnt/Wg and the receptors Wls were upregulated in L5, subsequently inhibiting the expression of intracellular Apc and Axin (Fig. 3C). Consequently, the key transcription factor *Arm* was upregulated in the leg tissue of L5 (Fig. 3C). To verify the indispensable role of the Wnt signaling pathway in ventral enlargement, we conducted RNAi of *Arm*, the key effector in this pathway, from the onset of L2 by two injections of *Arm* double-stranded RNA (ds*Arm*). As expected, the mRNA expression level of *Arm* was dramatically decreased in the ds*Arm* group, as detected by real-time quantitative PCR at day 3 postinjection (Fig. 3D), suggesting successful silencing of *Arm*. Notably, a specific reduction in petal-like femoral lobes was observed in the mid- and hind legs after the next molt (Fig. 3E). Further transcriptomic analysis showed that significantly differentially expressed genes (DEGs) were enriched in structural constituents of the cuticle and the Wnt signaling pathway (Fig. 3F), with *Cuticle* genes significantly downregulated in the ds*Arm* group (Fig. 3G), suggesting that the Wnt signaling pathway drives the ventral flat expansion by enhancing Cuticle expression (Fig. 3H).

Overall, these results suggested that the cuticle was the major structural constituent of the petal-like femoral lobes, and the enlargement in the ventral femur exoskeleton was regulated by Wnt signaling and the prolonged expression of Ubx in the early development stage.

## Ecdysone-dependent sexual dimorphism in body size

Sexually dimorphic traits are important for sexual selection and species survival[36,37]. The body size of the female orchid mantis is much larger than that of the male, representing a spectacular case of sexual

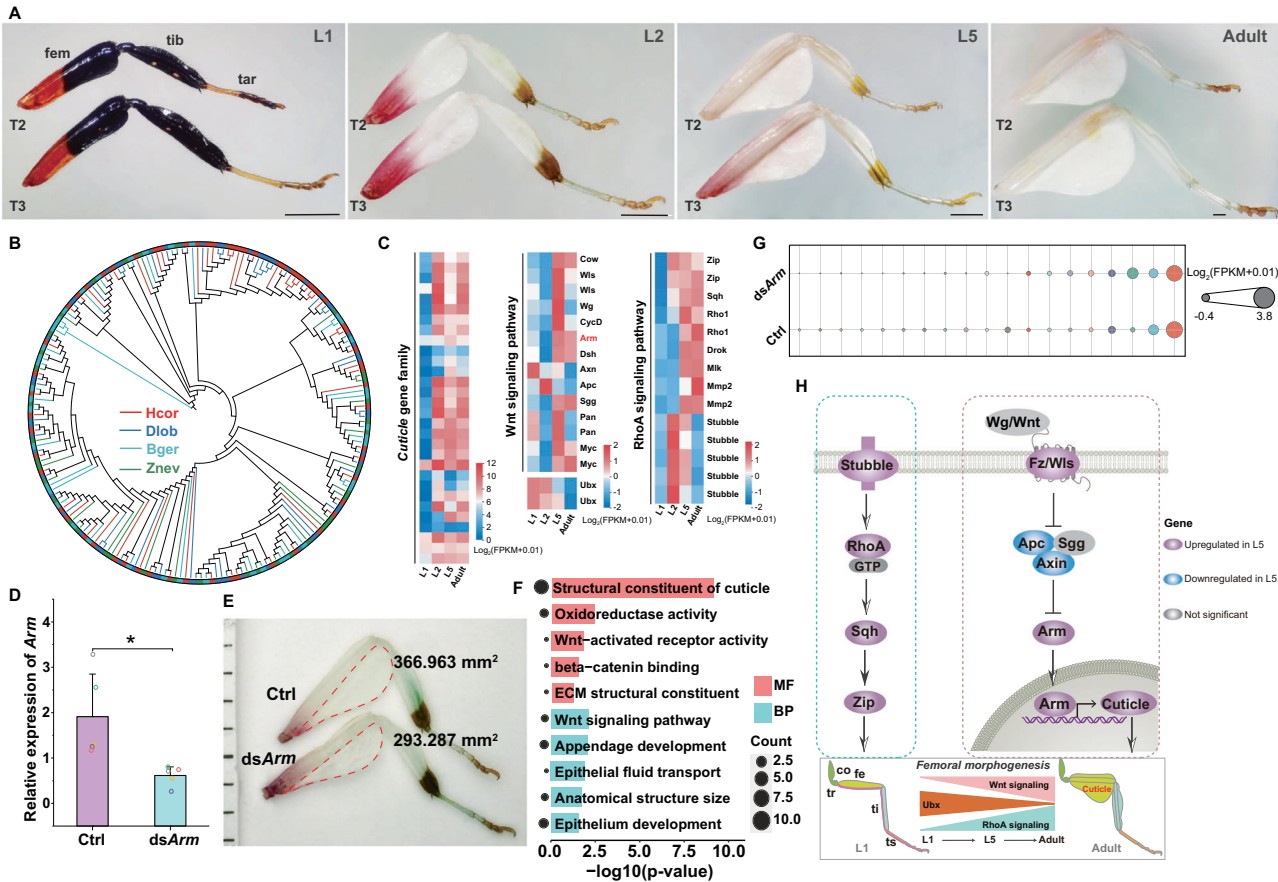

**Fig. 3 | Molecular mechanism of femoral morphogenesis for plant-mimicking camouflage. A** Photos of T2 and T3 legs across the developmental stage. Scale bar = 1 mm. **B** Gene family analysis showing that the *Cuticle* genes were significantly expanded in the genomes of *H. coronatus* and *D. lobata*. **C** Heatmaps of the differentially expressed genes in the leg tissues across the developmental stages. The red and blue colors indicate high and low expression levels, respectively.
**D** Expression level analysis of the *Arm* gene at day 3 after two injections of *Arm* double-stranded RNA (ds*Arm*). Ctrl, control group. Data represent the mean ± standard deviation of five replicates. *$p = 0.02$ by two-sided Student's *t* tests.
**E** Comparison of the size of 'petal-like' femoral lobes (the red dotted area) between the ds*Arm* and control groups. The numbers indicate the areas of these expansions.

**F** Gene Ontology (GO) enrichment of differentially expressed genes with FDR-corrected *p* value < 0.05 and |log2-fold-change| > 1 in the legs between the ds*Arm* and control groups. The enriched GO terms with corrected $p < 0.05$ determined by Fisher's exact test are presented. The size of the circles represents the number of differentially expressed genes for each GO term. The red and cyan colors represent the molecular function and biological process categories, respectively. **G** Bubble plot showing the expression levels of cuticle mRNAs between the ds*Arm* and control groups. **H** Reconstruction of gene regulatory pathways related to femoral remodeling. The red and blue colors represent significantly up- and downregulated genes in the leg tissues of the fifth-instar larvae compared to the first-instar larvae. co, coxa. tr, trochanter. fe, femur. ti, tibia. ts, tarsus.

dimorphism (Fig. 4A). Female orchid mantises and dead leaf mantises usually undergo 2–3 more molts, and as adults, are 60-65 mm long, i.e., 2–3 times larger than males (25-30 mm) (Fig. 4B). Compared to that in a close relative of mantises, *B. germanica*, which usually requires 5 instars for both sexes[38], the dimorphism in mantises was unusually extreme (Fig. 4B). The predominant explanation is that increased female size can increase fecundity and facilitate attracting more prey, whereas the small and mobile male facilitates mate-finding and reproductive success[18]. To investigate this regulatory mechanism, comparative transcriptomic analysis was performed between female and male adults of *H. coronatus* (Supplementary Fig. 15). We observed 1,892 downregulated and 1,858 upregulated transcripts in female mantises (Benjamini–Hochberg-corrected $p < 0.05$), most of which were enriched in the Hippo signaling pathway and insect hormone biosynthesis (Fig. 4C). As expected, sex steroid hormone biosynthesis was also significantly enriched between the two sexes.

The Hippo pathway is a key regulator of tissue size and an evolutionarily conserved signal regulating numerous biological processes, including cell growth, organ size control, and regeneration[39]. The insect hormone biosynthesis pathway controls the growth, development, metamorphosis, and reproduction of insects[40]. Gene set

enrichment analysis (GSEA) revealed that genes representing the Hippo signaling pathway, including the major effector *Yki*, were significantly upregulated in females (Fig. 4D). In addition, genes related to the synthesis of the ecdysteroid 20-hydroxyecdysone (20E), such as *Nvd*, Spo, *Phm*, *Dib*, *Sad* and *Shd*, were highly expressed in females (Fig. 4D, E). A previous study proved that the *Tai* gene is controlled by 20E, which can interact with the Hippo transcriptional coactivator *Yki* to promote the expression of canonical *Yki*-targeting genes[41]. Consistently, the responsive genes of *Yki*, such as *Diap1*, *Dally*, *Dmyc*, *Fj*, *Ex*, and *Crb*, were highly expressed in females (Fig. 4E), suggesting a potential role of ecdysis development in the induction of sexual dimorphism in body size of mantises through crosstalk between 20E and Hippo signaling.

## Expansion of the *Trypsin*, *CYP450*, and *UGT* gene families for adaptation to insectivory

Praying mantises are charismatic predators exclusively feeding on insects that are rich in protein, whereas close relatives are omnivorous, such as cockroaches, or herbivorous, such as termites, stick insects and crickets (Fig. 5A). To uncover the key genes involved in insectivory adaptation, comparative genomic analysis was performed, which

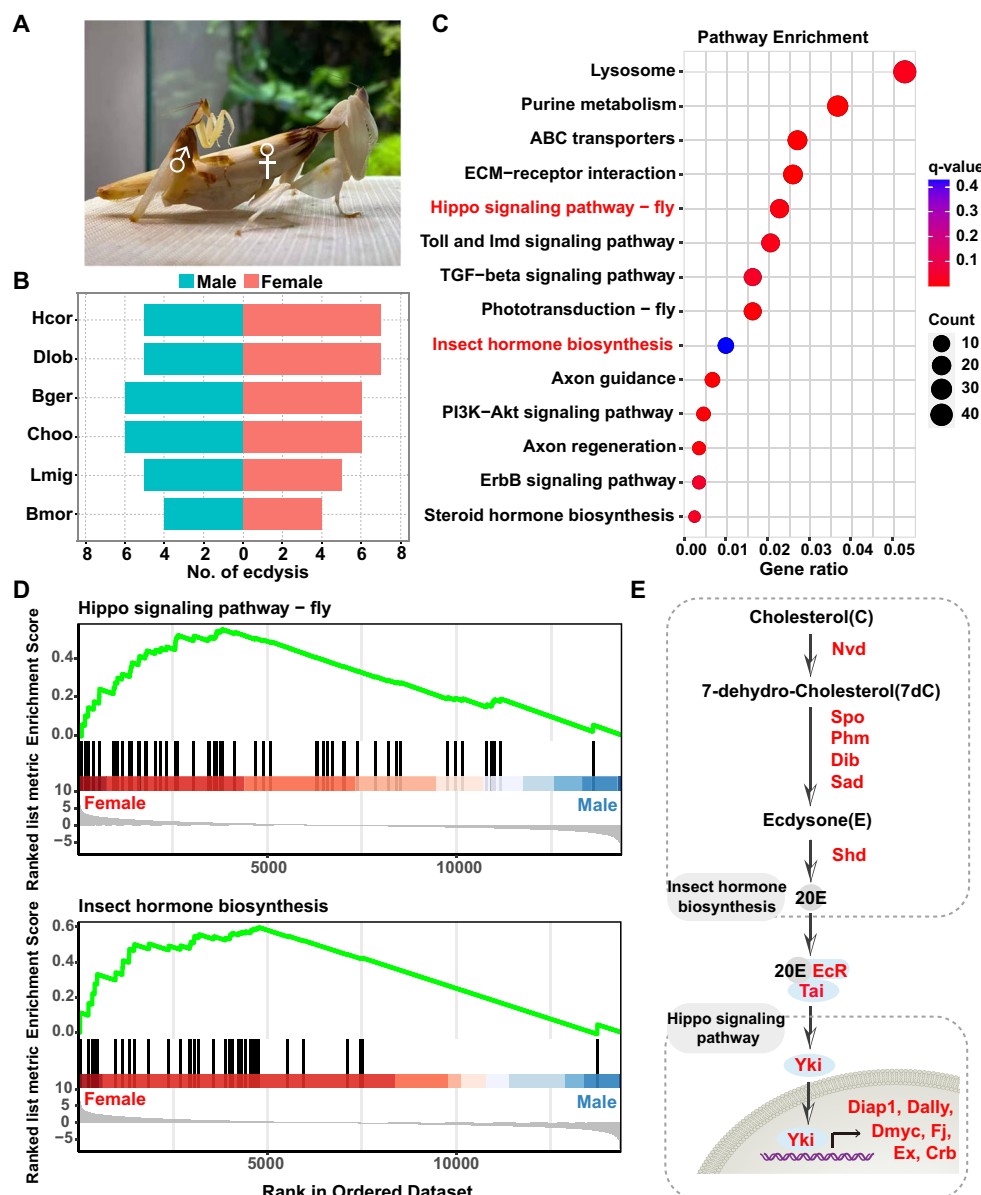

**Fig. 4 | Sexual size dimorphism and ecdysis development of the praying mantis.**
**A** Comparison of body size between male and female adults. **B** Lifetime average number of molts in both sexes showing sexual distinction in molting development of the praying mantises. The red and cyan colors represent females and males, respectively. The data graphed for Bger (*B. germanica*), Choo (*Clitarchus hookeri*), Lmig (*Locusta migratoria*), and Bmor (*Bombyx mori*) are from the literature[68–71]. **C** KEGG pathway enrichment of differentially expressed genes with FDR-corrected

*p* value < 0.05 and |log2-fold-change | > 1 between male and female orchid mantises. The size of the circle indicates the number of genes. The red and blue colors indicate low and high Benjamini-Hochberg corrected *p* values, respectively. **D** Gene set enrichment analysis (GSEA) showing the correlations of the genes involved in the Hippo signaling pathway and insect hormone biosynthesis with sex. **E** Crosstalk between ecdysone synthesis and the Hippo signaling pathway induces sexual size dimorphism in mantis. The red font represents the upregulated genes in females.

revealed significant expansion of *Trypsin* in *H. coronatus* and *D. lobata*, with 17 copies tandemly arranged on chromosome 18 of *H. coronatus* and chromosome 6 of *D. lobata* (Fig. 5B, C). Trypsin is a serine protease that breaks polypeptide chains into smaller peptides, facilitating dietary protein degradation[42]. We found that all the *Trypsin* genes were specifically expressed in the abdomen rather than in the head, thorax, or leg tissues (Fig. 5D). To further prove the correlation between *Trypsin* and insectivory, the enzyme activity of intestinal Trypsin of *H. coronatus* was determined by chemiluminescence with an herbivorous cricket as a control. Significantly higher enzyme activity of Trypsin was found in *H. coronatus* than in the cricket (Fig. 5E), further indicating the important role of *Trypsin* in insectivory adaptation in mantises.

As insectivores, mantises may frequently encounter poisonous prey, resulting in the accumulation of biotoxins and other xenobiotic

matter. As a consequence, the *cytochrome P450* (*CYP450*) and *UDP-glucuronosyltransferase (UGT)* gene families, especially *Cyp9e2* and *Ugt2b*, which are known to be involved in xenobiotic metabolism, underwent significant expansion in the genomes of *H. coronatus* and *D. lobata* (Supplementary Fig. 16). These results suggest that expansion of the *Trypsin, CYP450*, and *UGT* gene families plays an important role in adaptation to an insectivorous diet.

Due to their exceptional plant-mimicking phenotypes and suite of evolutionary adaptations, mantises can serve as an outstanding model for studying complex phenotypic innovation and adaptive evolution. Reference genomes and accurate annotations are critical for evolutionary, developmental, and functional analyses[43]. Our study presents two reference genomes and gene catalogs for the Mantodea and reveals the genetic and biochemical bases of plant-mimicking

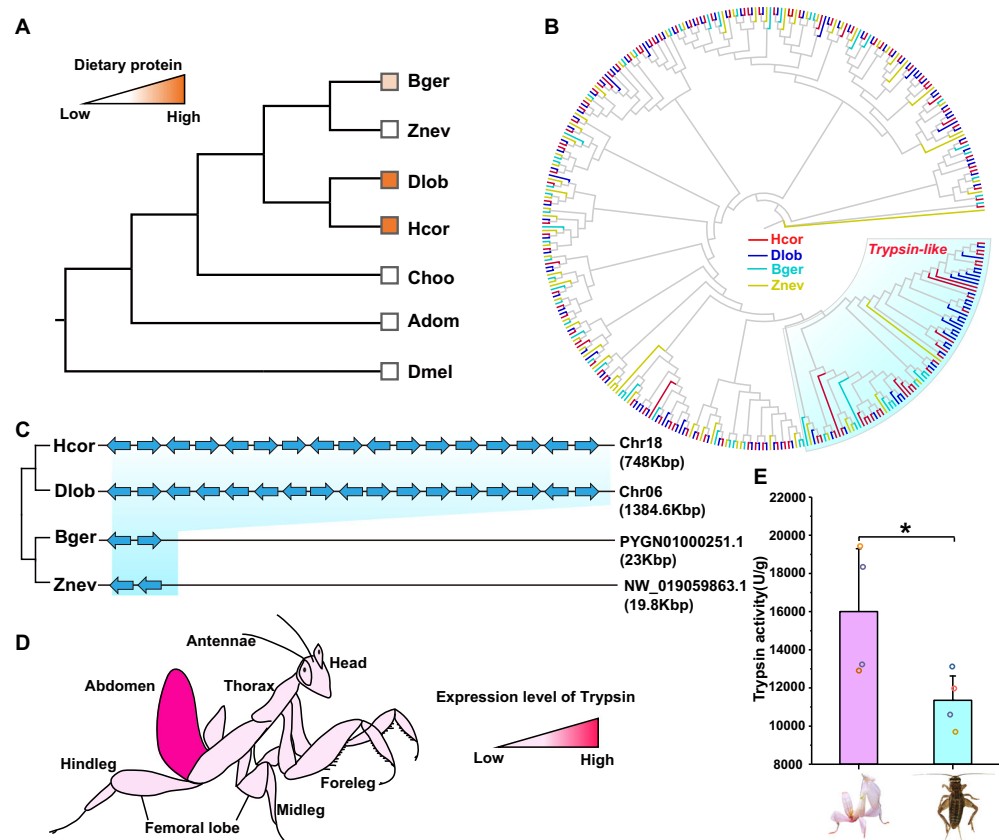

**Fig. 5 | Evolution of the *Trypsin* gene family for insectivory adaptation in mantises. A** Comparison of dietary protein content between praying mantises and other insects. **B** Phylogenetic analysis showing that the *Trypsin* gene family was significantly expanded in *H. coronatus* and *D. lobata*. **C** Schematic representation of the tandem replication of the *Trypsin* gene cluster in praying mantises. Each arrow indicates a complete gene oriented in the 5′ to 3′ direction. **D** A color diagram showing the expression level of Trypsin in the head, thorax, leg, and abdomen tissues. **E** Detection of trypsin enzyme activity in the intestines of a carnivorous mantis and an herbivorous cricket. Data represent the mean ± standard deviation of four replicates. * $p = 0.04$ by two-sided Student's *t* tests.

coloration and femoral appendages in the praying mantis by comparative genomic and transcriptomic analyses in combination with functional validation. Moreover, we found evidence to support the molecular basis of ecdysone-dependent sexually dimorphic body size and adaptation to an insectivorous diet. In conclusion, our study provides valuable genomic resources, as well as evolutionary developmental insights into camouflage phenotypic innovation and adaptation in arthropods.

## Methods

### Sample collection

Captive breeding individuals of *H. coronatus* (Mantodea, Hymenopodidae) hatched from the same ootheca that was collected from the Xishuangbanna rainforest, Yunnan Province, China in 2018. Individuals of *D. lobata* (Mantodea, Deroplatyidae) were collected from a captive breeding center in Beijing, China in 2018. All individuals were housed in semitransparent cages (7 cm × 7 cm × 9 cm) in an incubator with a temperature range of 26–27 °C, 80% relative humidity, and a 16 h/8 h light/dark photoperiod. Larvae from the first to third instars were fed fruit flies and switched to crickets from the fourth instar onward. To ensure sufficient output of nucleic acids for Nanopore sequencing and Illumina paired-end sequencing, adult female *H. coronatus and D. lobata* were used for nucleic acid preps. DNA was extracted using the DNeasy Blood and Tissue Kit following the manufacturer's protocol (Qiagen, Valencia, CA, USA). The quantity and quality of DNA were checked by a Qubit 2.0 (Thermo Fisher Scientific) and agarose gel electrophoresis, respectively. RNA was extracted using TRIzol reagent (Thermo Fisher Scientific).

All sample collections and animal experimental procedures were approved by the Institutional Animal Care and Use Committee of the Institute of Zoology, Chinese Academy of Sciences. All efforts were made to minimize the suffering of the animals.

### Genome sequencing, Hi-C sequencing, and genome assembly

A combination of Nanopore sequencing, Illumina sequencing and Hi-C sequencing was used to generate mantis genome assemblies. Paired-end libraries with insert sizes of 400 bp were constructed and sequenced using the Illumina HiSeq X-Ten platform following the standard manufacturer's protocol (San Diego, USA), yielding 155.6 Gb of clean reads for *H. coronatus* and 286.3 Gb for *D. lobata*, followed by quality filtering (Supplementary Table 9). The high-quality reads were used for genome size estimation by the K-mer method. 1D libraries were constructed according to the manufacturer's instructions (SQK-LSK109, Oxford Nanopore Technologies, UK) and sequenced on the Nanopore PromethION platform (Oxford Nanopore Technologies, UK). In total, 297.60 Gb of subreads for *H. coronatus* and 258.05 Gb for *D. lobata* were generated with a high sequencing depth (approximately, 66–103×) (Supplementary Table 9). To further improve the continuity of the assembled genomes and anchor the assemblies into chromosomes, Hi-C sequencing was performed to order and orient the contigs, as well as to correct misjoined sections and merge overlaps. To construct Hi-C libraries, tissue samples were fixed immediately in formaldehyde solution once isolated. DNA was digested with the restriction enzyme DpnII overnight. Hi-C libraries with a mean size of 350 bp were constructed using NEBNext Ultra enzymes and Illumina-compatible adaptors and sequenced using the Illumina HiSeq 2500

platform. All libraries were quantified by a Qubit 2.0 (Thermo Fisher Scientific), and insert size was checked using an Agilent 2100.

Contigs were assembled by NextDenovo software (v2.0-beta.1, https://github.com/Nextomics/NextDenovo), and the assembled contig-level genomes were polished by NextPolish (v1.0.5)[44] and Pilon (v1.22)[45]. Then, the contigs were anchored into chromosomes by Hi-C sequencing reads through the Juicer (v1.5)[46] and 3D-DNA (v180922)[47] software workflows. To further improve the chromosome-scale assembly, it was subjected to manual review and refinement using Juicebox Assembly Tools (https://github.com/theaidenlab/juicebox). Finally, genome quality was estimated with BUSCO (insecta_odb9, v3.0.2)[48] and k-mer analysis and by mapping the initial reads back to the assembly.

### Transcriptome sequencing and analysis

RNA from whole individuals of *H. coronatus* (first-, second-, and fifth-instar larvae and adults) was used for Nanopore full-length cDNA sequencing. Meanwhile, samples of the brain, thorax, abdomen, and mid and hind legs of each individual were sequenced using Illumina HiSeq 2500 sequencing. Fifth-instar larvae of *D. lobata* were used for Nanopore full-length cDNA sequencing and Illumina HiSeq 2500 sequencing. RNA from female and male subadults was extracted and subjected to Illumina HiSeq 2500 sequencing to investigate sex-related DEGs. At least three replicates were used for each group.

For *H. coronatus*, 222.31 Gb of Nanopore full-length cDNA sequencing data and 573.28 Gb of Illumina read cDNA sequencing data were obtained across different developmental stages and assembled into transcript datasets, yielding 13,817 transcripts (Supplementary Table 9). For *D. lobata*, 123.60 Gb of Nanopore full-length cDNA sequencing data and 68.21 Gb of Illumina read cDNA sequencing data were obtained from adult individuals and yielded 15,494 transcripts (Supplementary Table 9).

Relative gene expression was measured as transcripts per million reads, and DEGs were identified using DESeq2 (v1.0)[49] with false discovery rate (FDR) corrected p value (q-value) set to less than 0.05 and |log2-fold-change| set to greater than 1. Gene ontology functional enrichment and KEGG pathway analysis of the DEGs were performed with Goatools and KOBAS[50], respectively, with a Benjamini–Hochberg-corrected p value less than 0.05 indicating statistical significance.

### Genome annotation

Transposable elements were identified using RepeatMasker (open-4.0.7), RepeatModeler (v1.0.8), and MITE-Hunter (v1.0.8)[51]. Gene structures were determined by combining ab initio and homology methods. For ab initio annotation, we used Augustus (v3.2.1)[52] and GENSCAN (v1.0)[53] to analyze the repeat-masked genome. For homolog-based annotation, protein sequences of the fruit fly (*Drosophila melanogaster*), cockroach (*Blattella germanica*), honeybee (*Apis mellifera*), mosquito (*Aedes aegypti*), and small brown planthopper (*Laodelphax striatella*) genomes were aligned to mantis genome sequences using BLAST software (v2.3.0)[54]. Together with transcriptomic data, gene sets from these three methods were then integrated by EVidenceModeler software (v1.1.1)[55]. For gene functional annotation, the integrated gene set was aligned against public databases, including KEGG, SwissProt, TrEMBL, COG, and NR, with BLAST (v2.3.0)[54] and merged with annotations by InterProScan (v4.8) software[56]. Annotation integrity was estimated by comparison with reference genome annotations and BUSCO (v3.0.2)[48], resulting in 95.5–98.2% completeness according to BUSCO analysis, suggesting the high quality of the annotation.

### Identification of orthologous groups and phylogenomic analysis

To cluster families of protein-coding genes, we extracted protein sequences from the genomes of *H. coronatus*, *D. lobata*, and 16 other species of Insecta: *Zootermopsis nevadensis* (Blattodea: Termopsidae), *B. germanica* (Blattodea: Ectobiidae), *D. melanogaster* (Diptera: Drosophilidae), *Locusta migratoria* (Orthoptera: Acrididae), *Clitarchus hookeri* (Phasmatodea: Phasmatidae), *Bombyx mori* (Lepidoptera: Bombycidae), *A. mellifera* (Hymenoptera: Apidae), *Ctenocephalides felis* (Siphonaptera: Pulicidae), *Stenopsyche tienmushanensis* (Trichoptera: Stenopsychidae), *Folsomia candida* (Entomobryomorpha: Isotomidae), *Ladona fulva* (Odonata: Libellulidae), *Pediculus humanus* (Phthiraptera: Pediculidae), *Photinus pyralis* (Coleoptera: Lampyridae), *Campodea augens* (Diplura: Campodeidae), *Rhopalosiphum maidis* (Hemiptera: Aphididae), and *Ephemera danica* (Ephemeroptera: Ephemeridae). Protein sequences showing redundancy caused by alternative splicing variations or premature codons were removed from the protein-coding genes. The protein sequences were aligned reciprocally (i.e., all-vs.-all) using BLASTP programs with an E-value ≤ 1e − 5 and then clustered using OrthoMCL (v2.0.9)[57]. Finally, 22,455 orthogroups were identified among the 18 species.

To reveal the phylogenetic relationships among Mantodea and other insects, the protein sequences of 324 1:1 orthologs from all 18 species were aligned with MUSCLE (v3.8.31)[58] and then concatenated using in-house Perl scripts. RAxML (v8.2.10)[59] was used to construct a phylogenetic tree for the superalignment using the GTRGAMMA model with *Campodea augens* and *Folsomia candida* as outgroups. We used the best-fitting substitution model as deduced by ProteinModelSelection.pl and 1000 replicates for bootstrap support. The MCMCTree program of the PAML (v4.8)[60] package was used to determine divergence times with the approximate likelihood calculation method and three dated fossil records (*Baissatermes lapideus* (145–99.6 Ma), *Raphogla rubra* (260–251 Ma), and *Rhyniella praecursor* (412.3–391.9 Ma)) from TimeTree (http://www.timetree).

### Expansion and contraction of gene families

Family expansion/contraction analysis was performed by CAFÉ (v3.1)[61] calculations with the parameters lambda -s and p < 0.01 based on the phylogenetic tree constructed above. Gene expansion and contraction results for each branch of the phylogenetic tree were obtained.

### Demographic history reconstruction

To trace the demographic history of *H. coronatus* and *D. lobata*, we employed PSMC to estimate changes in effective population size using heterozygous sites[25], with the following set of parameters: −N 30 −t 15 −r 5 −p 4 + 25 × 2 + 4 + 6. *H. coronatus* and *D. lobata* take approximately 8 months to reach adulthood and then mature sexually after 25–35 days, so the generation time (g) was set to 0.75 years. The nucleotide mutation rate (μ) of *H. coronatus* was estimated to be $1.85 \times 10^{-9}$ mutations per site per generation with *D. lobata* as the reference species for comparison using the following formula: $\mu = D \times g/2T$[62], where D is the observed frequency of pairwise differences between the two species, T is the estimated divergence time, and g is the estimated generation time for *H. coronatus* and *D. lobata*. Global mean surface temperature was estimated from benthic $d^{18}O$[63].

### Genome duplication

To investigate the genome evolution of *H. coronatus* and *D. lobata*, we searched for genome-wide duplications with *B. germanica* as an outgroup. We identified different modes of gene duplication as whole-genome duplications (WGDs), tandem duplicates (TDs), proximal duplicates (PDs, gene distance on the same chromosome less than 10), transposed duplicates (TRDs, transposed gene duplications), or dispersed duplicates (DSDs) using DupGen_finder[64] with default parameters. Five duplication categories were identified in the *H. coronatus* and *D. lobata* genomes, including DSD (53.59%), TD (23.12%), TRD (14.12%), PD (8.42%), and WGD (0.76%). Then, the overlap between the various types of duplicated genes and expanded genes was identified, yielding 796 tandem replication genes belonging to expanded gene families, including the *Cuticle* gene family.

To verify whether the identified tandem gene clusters existed, we mapped the Illumina and Nanopore reads back to these two genome assemblies and examined whether the sequencing depth of each tandem replicated gene was consistent with the sequencing depth of the whole genome. Finally, a uniform sequencing depth of all the duplicated genes consistent with the sequence depth of the whole genome was observed, confirming that these tandem gene clusters were real.

## Expansion of the *ABC, Cuticle, Trypsin, CYP450*, and *UGT* gene families

Genome-wide protein sequences of *H. coronatus*, *D. lobata*, *Z. nevadensis*, *B. germanica*, and *D. melanogaster* were extracted, and the conserved nucleotide binding domain (PF00005.24) and transmembrane domain (PF00664.20) were scanned genome-wide for candidate ABC transporter genes using the Hidden Markov Model (HMM) in R (v3.2.1)[52]. To assign the candidate ABC genes to different subfamilies, multiple alignments of the ABC transporter protein sequences were performed using MUSCLE (v3.8.31)[58], and the poorly aligned regions and partial gaps were removed with trimAI (gt = 0.5). Then, the alignments were subjected to phylogenetic analysis using RAxML based on the maximum likelihood method with the following parameters: -f a -x 12345 -N 1000 -p 12345 -m PROTGAMMAJTTX. The resulting trees were displayed and edited using FigTree (v1.4.4, https://github.com/rambaut/figtree/releases). The subfamily assignment of ABC proteins in each species was further confirmed using BLASTP analyses on the NCBI webserver (https://www.ncbi.nlm.nih.gov/blast). In addition, the same analyses were performed to identify *Cuticle* (pfam:PF00379), *Trypsin* (pfam:PF00089), *UGT* (pfam:PF00201), and *CYP450* (pfam:PF00067). We also performed manual annotation of these genes to avoid omissions in their general feature format (GFF) files.

## Extraction of insect pigments and color reaction in vitro

To extract the pigment of the orchid mantises, second-instar individuals were crushed in liquid nitrogen and dissolved in acidified methanol (hydrochloric acid at 0.5%)[65], followed by incubation in a thermostatic shaker for 48 h at 220 rpm and 25 °C. After centrifugation, the supernatant was collected and passed through a 0.22-μm filter. The redox-dependent color changes of the pigments were observed by adding 10 μL of oxidant (NaNO$_2$ with a concentration of 1%) and reductant (ascorbic acid with a concentration of 1%). In acidified methanol, the redox states of the ommochrome pigments were stable, and they were stored at −20 °C. Five individuals were assigned to each group.

## Transmission electron microscopy

The ultrastructure of the mid and hind legs was investigated by transmission electron microscopy. The legs were dissected from fifth-instar *H. coronatus* and fixed with 3% glutaraldehyde in 0.2 mol/L phosphate buffer (pH 7.2) for 48 h at 4 °C. The legs were then rinsed 3 times with phosphate buffer followed by postfixation in 1% osmium tetroxide for 3 h at 4 °C. The legs were washed several times, for 10 min each time, and placed into a series of ascending concentrations of acetone (50, 70, 80, 90, and 100%) for dehydration. The samples were embedded in Epon 812 for 2 h at room temperature. Then, the samples were trimmed to prepare ultrathin sections. The ultrastructure was captured by a JEM-1200EX transmission electron microscope (TEM, JEOL, Japan).

## High-performance liquid chromatography analysis

Individuals of the first, second, and fifth instars were used to determine the concentrations of kynurenine and 3-hydroxykynurenine. The same amount of leg tissue was homogenized directly, added to a methanol solution containing the internal standard, homogenized again, and then centrifuged at 4 °C and 18,000 × *g* for 20 mins (Microfuge 20 R, Beckman Coulter, Inc., Indianapolis, IN, USA). The supernatant was used to detect the concentrations of kynurenine and 3-hydroxykynurenine

according to the protocol of a previous study[66]. The standards of kynurenine (CAS No. 343-65-7) and 3-hydroxykynurenine (CAS No. 484-78-6) were purchased from Sigma–Aldrich. Five individuals were assigned to each group.

## Histological analysis of the petal-like femoral lobes and size measurement of leg tissue

Hematoxylin-eosin staining was performed for structural observation of the leg tissue of *H. coronatus*. Briefly, the mid and hind legs of *H. coronatus* were fixed in 4% paraformaldehyde overnight at 4 °C. Then, paraffin-embedded, 4 μm tissue slices were cut and placed on slides for hematoxylin-eosin staining, with hematoxylin staining the nucleus and eosin staining the cytoplasm.

ImageJ2 (http://imagej.net/) was used to measure the length and area of the femur, tibia and tarsus of T2 and T3 legs of *H. coronatus*. Briefly, each segment of leg was dissected and taken photos. The scale bar of image was set followed by marking the target region by color threshold to automatically measure the area. The length of the target was measured by straight line.

## RNA interference

To validate the function of *Arm* in the morphological development of legs, double-stranded RNAs (dsRNAs) were synthesized for each gene. The dsRNA sequences are shown in Supplementary Table 10. dsRNA targeting each gene was injected into the right midleg of first-instar individuals at 0.1 μL (660 ng/μL) using a microinjection system, and the second injection was conducted as described above at day 5 after the first injection. Control group individuals were injected with the same amount of PBS. Four individuals were assigned to each group. The mRNA expression levels of these genes were detected at day 3 after the second injection. One individual per group was cultured to molting for observation of the phenotypic outcome.

## Real-time quantitative polymerase chain reaction (PCR)

Total RNA was extracted from the femur, tibia and tarsus of T2 and T3 legs using RNAprep Pure Micro Kit (DP420, TIANGEN Biotech (Beijing) Co.,Ltd). RNA quality was determined using a NanoDrop 2000 (Thermo Fisher Scientific Inc.), and RNA of suitable quantity was reverse transcribed using a PrimeScript RT Reagent Kit (RR037A, TaKaRa Biotechnology Co., Ltd.). Quantitative real-time PCR was performed using TB Green Premix Ex Taq II (Tli RNaseH Plus) (RR820A, TaKaRa Biotechnology Co., Ltd.) on an Mx3000P Real-Time PCR System (Agilent Technologies, Inc.). The relative expression level of the *Ubx* and *Arm* genes was quantified using *GAPDH* as the internal reference. Relative fold-change in gene expression was calculated using the delta-delta cycle threshold method[67]. Three biological repeats were assigned to each group. Genes with a |fold-change| of greater than 2 and an adjusted *p* value less than 0.05 were identified as DEGs.

## Enzyme activity of trypsin

Insect guts were isolated for Trypsin detection. Samples were immediately frozen in liquid nitrogen and stored at −80 °C before use. The enzyme activity of trypsin was detected using a commercial kit from Solarbio Technology Company (BC2315; Beijing, China). In this experiment, N-benzoyl-L-arginine-ethylester (BAEE) was used as the substrate. Under the catalysis of Trypsin, the ester bond of BAEE was hydrolyzed to produce a molecule of N-benzoyl-L-arginine (BA) and ethanol. The UV absorption of BA at 253 nm was much higher than that of BAEE. The amount of enzyme that increased by 0.001 per minute was calculated as one unit of activity. Four individuals were assigned to each group.

## Statistical analysis

Measurements of continuous variables are expressed as the means ± SDs. Unless otherwise stated, differences between two groups were

assessed using Student's *t* tests, and differences among three or more groups were assessed using one-way analyses of variance in SPSS v18 (SPSS). Dunnett's post hoc tests were used to compare treatment groups to controls. Adjusted $p < 0.05$ was considered statistically significant.

## Reporting summary
Further information on research design is available in the Nature Portfolio Reporting Summary linked to this article.

## Data availability
The sequence data and the genome assemblies have been deposited to the National Genomics Data Center, China (https://ngdc.cncb.ac.cn/), with accession number CRA010804 (https://bigd.big.ac.cn/gsa/browse/CRA010804). Source data are provided as a Source Data file. Source data are provided with this paper.

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

## Acknowledgements

This study is supported by the Strategic Priority Research Program of Chinese Academy of Sciences (XDB31000000, Fuwen Wei), the National Natural Science Foundation of China (31821001, Fuwen Wei), and the Youth Innovation Promotion Association, CAS (2023090, Guangping Huang). We thank Hao Wang and Chaotai Wei for providing the ecological photos of mantises, thank Zongyi Sun for help with genome annotation amending, and thank Dr Yang Li and Chuang Gao for help with RNA interference.

## Author contributions

F.W. conceived and supervised this study. G.H. performed the sample collection. G.H. and L.S. conducted data analysis and interpretation. L.S. and G.H. performed the gene family analysis with input from X.D. and X.H. L.S. and G.H. conducted the functional experiments. G.H., L.S. and F.W. discussed the results and wrote the paper. All authors have read and approved the final manuscript.

## Competing interests

The authors declare no competing interests.
