## [Peer Review File · Nature Communications]

Evolutionary genomics of camouflage innovation in the orchid mantisREVIEWER COMMENTS

Reviewer #1 (Remarks to the Author):

Camouflage phenomena are widespread in nature, and the orchid mantis is a very representative and charismatic example of camouflage. Huang et al. not only provide an important reference genomic resource for the plant-mimicking mantids, but also explore the genetic mechanisms of multiple adaptive traits in the orchid mantis such as camouflage coloration, morphological specialization, sexual dimorphism, and insectivory adaptation, providing an important foundation and idea for subsequent studies. I think the important contribution of this paper is that it gives the orchid mantis the potential to be studied in depth as a model. Overall the paper feels very complete and tells an impressive story, which would attract the attention of a general biology readership. I thus recommend that this paper be accepted with minor revisions.

L76–87: This paragraph seems to be the summary of studying the evolution and genetic mechanisms of both *H. coronatus* and *D. lobata*, whereas for the genetic mechanisms, this paper focuses on the orchid mantis *H. coronatus*. I suggest that the results of this study be clearly summarized here.

L161: Please provide a reference for the PSMC method.

L154–159: I suggest a brief description of the habitat and current distribution of these mantids, which will help the readers to understand their diversification and ecological expansion.

L249–251: It will be helpful to provide a heatmap including transcriptome-wide differential expression patterns as a supplementary figure, which will give the readers an overall impression of the expression differences across developmental stages. An alternative way is to provide volcano plots showing differentially expressed patterns with the Cuticle genes and Ubx genes labeled, which will help to enhance the logic from transcriptome data to specific gene expression patterns.

L314–315: If possible, please also provide a heatmap or volcano plots to illustrate the transcriptome-wide differential expression patterns between females and males with the Yki gene and other focal genes labeled.

L498: Please specify the three dated fossil records used for tree calibration.

L656: Please provide SRA accession numbers for the raw reads and the genome assemblies.

Fig. 1: I suggest adding ecological photos of these praying mantids to this figure, which will help the readers visualize and understand their adaptive camouflage.

Extended Data Figures: I noticed that the abbreviations of species names in some figure legends are italicized as well as the species names, but they are not italicized in the figures, so it is recommended to keep the figure legends and figures consistent.

Extended Data Fig. 10: The internal nodes of the phylogenetic tree have different colors. Please specify their meanings.

Reviewer #2 (Remarks to the Author):

Praying mantises (referred to as insect order Mantodea) represent a predatory polyneopteran insect lineage for which a reference genome has not been available so far. A de novo assembly of two mantid genomes has now been provided in the study submitted by Huang et al. Based on two Southeast Asian taxa, the orchid mantis *Hymenopus coronatus* (Hymenopodinae) and the dead leaf imitator *Deroplatys lobata* (Deroplatyidae, not Mantidae as stated by the authors). Both species exhibit an unusual, specialised morphology among praying mantises and indeed can be considered highly charismatic taxa. Based on the genomic data assembled the authors infer the potential developmental genetic base of leg enlargements and colouration that largely contribute to the animals' stunning plant mimicry. Technically there are no obvious flaws or unjustified conclusions (but I am no expert in developmental biology or bioinformatics), but the authors tend to overstate their findings. The de novo assembly of genomes for understudied organismic groups undoubtedly is a worthwhile endeavour per se, however these kind of studies are frequently published in genome report journals and no longer justify publication in high-ranking journals unless the conclusions are of major significance for our understanding of biological phenomena in general. Here, one conclusion (based on transcriptomes/gene expressions of various instars) is that the cuticle is responsible for the leaf-like expansions of the legs and that Wnt signaling determined by Ubx is involved in this process, two observations of interest but not unexpected. To satisfactorily corroborate this, it would have been favourable to compare the results with one of the numerous mantid species that does not exhibit leaf-like leg expansions (as a control!). The two species examined in fact are rather closely related and highly subordinate within the mantid tree of life. Investigation of less related taxa stemming from different major mantid lineages and different geographic regions would also have allowed for more general statements in regard of genomic properties, e.g. gene family expansions/contractions; Trypsin gene family expansion in mantises. These statements cannot be made for all mantises but only for the rather derived narrow lineage that is represented by *Hymenopus* and *Deroplatys*. This is not only a deficiency for statements regarding mantids but for insects in general (Figure 1) when only a single cockroach, a single termite, a single dragonfly etc. is included (it is also difficult to see which insect orders are represented therein). Furthermore, the statement that "the lack of a whole-genome sequence for any Mantodea species has impeded the resolution of the phylogenetic position of this group within Insecta" is simply not true: The 324 single-copy orthologs used for reconstructing the phylogeny of 18 insect species in the present study pales in comparison to the 1478 single-copy nuclear genes used by Misof et al. (2014) for their phylogenomic study of over 100 insect species including mantises and allies. This study is cited and supported by the present data, so it is unclear what prompted the authors to make this statement. There are further conclusions that just overemphasise the significance of the study when the authors suggest based on the limited temporal demographic estimations "that proper conservation plans should be developed for mantises in response to ongoing climate change" (This statement might just be true for most organisms on this planet anyway).

This said, there is valuable and noteworthy information provided here that should be published, but presented less fulsome and in a more cautious manner. In my opinion, this study is better suited for a less high-ranking and more specialised journal focussed on genome reports.

Reviewer #3 (Remarks to the Author):

In this study, Huang et al. focus on unique aspects of camouflage morphology in orchid mantids, mainly aposematic coloration at L1 stage and "petal-like" femoral enlargement in adults. To begin to understand mechanisms responsible for these phenotypic innovations, the authors first generate de novo assembly of two chromosome-level genomes (one for the orchid mantis, and one for its close relative, the dead leaf mantis). Then, by using comparative genomic analysis authors identify Scarlet gene as a key player for mantis camouflage coloration. Finally, functional testing via RNAi, reveals that the "petal-like" morphology is regulated by WNT signaling. Overall, this is excellent and very substantial study, which provides novel insights into the origins of morphological novelties in an exciting new model insect (orchid mantis). The quality and the amount of work is exceptional and text

is easy to read and follow. One issue that has to be corrected, though, is authors' simplistic interpretation of the role of Ubx at L1 stage and its putative role in inhibiting Wnt signalling (for details see below). Aside from that, this is indeed a very nice work that will be of broad interest to a large audience of evolutionary and developmental biologists.

Aleksandar Popadić

Main concern:

1) The authors present interpretation regarding the role of Ubx at L1 stage is likely incorrect, or at least greatly speculative. This is due to their reliance on several *Drosophila* papers (Diaz-de-la-Loza et al. 2020; Oberhofer et al. 2014), which are not relevant for hemimetabolous insects such as mantids. For that matter, the cited *Tribolium* paper (Lewis et al. 2000) is also inappropriate as it deals specifically with the development of A1 appendage, and not the T3 leg. Instead, the authors should get acquainted with now classic work that shows that Ubx regulates T3 leg enlargement in hemimetabolous insects (Mahfooz et al 2004; Mahfooz et al 2007). Alternatively, it can also enlarge T2 leg (Khila et al. 2009).

If the orchid mantis follows the general trend observed in other mantids, then its T3 tibia and tarsus are enlarged compared to their T2 counterparts (at L1 stage; see Mahfooz et al 2004). If observations in *Tenodera* apply to *Hymenopus* and *Deroplatys* then Ubx is expressed specifically in T3 tibia and tarsus, and not in the femur in the latter two species. In other words, the authors' entire explanation about the role of Ubx becomes false, as Ubx is not expressed in T3 femur. Consequently, the experimental evidence highlighted in the paper is misleading as it refers to Ubx mRNA level sampled from the entire leg (including tibia and tarsus).

The authors should do the following: repeat the experiment and use individual leg segments as a tissue source (T3 femur; T3 tibia; and T3 tarsus). Alternatively, they can perform antibody staining on *Hymenopus* embryos using a cross-reacting UbdA antibody (Kelsh et al 1994). Either would prove that Ubx is not localized in T3 femur, and hence, is not responsible for the absence of petal-like morphology at L1 stage.

Note: the femoral lateral extensions are not restricted to T3 leg only, but are present on all three leg pairs – this is another evidence that Ubx is not involved in these modifications as it is never expressed in T1 leg (and only very rarely in T2 leg).

2) In the light of the importance of the "petal-like" cuticular enlargement, a more detailed visual presentation should be provided. Authors should create a new Fig 1 that shows both the details of color differences between L1 and adult forms, as well as femoral modifications from L2-adult stage. This figure should encompass images of L1, L2, and adult stages. In addition, it should display dissected T2 and T3 legs in L1 (showing absence of "petal-like" morphology), then dissected T2 and T3 legs in L2 and adult stages (showing the presence of this trait).

Reviewer #1 (Remarks to the Author):

Camouflage phenomena are widespread in nature, and the orchid mantis is a very representative and charismatic example of camouflage. Huang et al. not only provide an important reference genomic resource for the plant-mimicking mantids, but also explore the genetic mechanisms of multiple adaptive traits in the orchid mantis such as camouflage coloration, morphological specialization, sexual dimorphism, and insectivory adaptation, providing an important foundation and idea for subsequent studies. I think the important contribution of this paper is that it gives the orchid mantis the potential to be studied in depth as a model. Overall the paper feels very complete and tells an impressive story, which would attract the attention of a general biology readership. I thus recommend that this paper be accepted with minor revisions.

Response: We are very grateful for your constructive comments and the time taken to review our manuscript. We have revised the manuscript throughout accordingly. Below is our point-to-point response to each of the comments.

L76–87: This paragraph seems to be the summary of studying the evolution and genetic mechanisms of both *H. coronatus* and *D. lobata*, whereas for the genetic mechanisms, this paper focuses on the orchid mantis *H. coronatus*. I suggest that the results of this study be clearly summarized here.

Response: Thanks for this very important suggestion. We have corrected this statement as below (lines 75-82).

“The lack of genomic resources for Mantodea lineages has impeded further in-depth molecular and developmental investigation. To address the genetics and evolutionary mechanism of plant-mimicking phenotypic innovation of *H. coronatus*, we de novo-assembled the chromosome-level genomes of the flower-mimicking species *H. coronatus* with dead leaf-mimicking species *D. lobata* as a contrast. Comparative genomic and developmental transcriptomic analyses, and genetic engineering experiments based on RNA interference (RNAi) were performed to dissect the genetic basis of plant-mimicking camouflage.”

L161: Please provide a reference for the PSMC method.

Response: We apologize for missing this citation, and the reference below has been added in the revised manuscript (line 163, reference #24).

Reference: Li H, Durbin R. Inference of human population history from individual whole-genome sequences. Nature. 2011, 475(7357):493-6.

L154–159: I suggest a brief description of the habitat and current distribution of these mantids, which will help the readers to understand their diversification and ecological expansion.

Response: Thanks for this insightful suggestion. The distribution of the orchid mantis has been reported in the rainforest across Malaysia, Indonesia, India, Thailand, Vietnam and Southern China, while that of the dead leaf mantis is mainly in the rainforest of Malaysia and Philippines. We have provided this description in the revised manuscript (lines 171-175). However, both the orchid mantis and dead leaf mantis are rarely encountered in the wild, so there is little information available on their microhabitat or fine scale distributions.

L249–251: It will be helpful to provide a heatmap including transcriptome-wide differential expression patterns as a supplementary figure, which will give the readers an overall impression of the expression differences across developmental stages. An alternative way is to provide volcano plots showing differentially expressed patterns with the Cuticle genes and Ubx genes labeled, which will help to enhance the logic from transcriptome data to specific gene expression patterns.

Response: Thanks for this very important suggestion. We have provided the heatmap of differential expression genes across developmental stages in Figure 3C.

L314–315: If possible, please also provide a heatmap or volcano plots to illustrate the transcriptome-wide differential expression patterns between females and males with the Yki gene and other focal genes labeled.

Response: Thanks. We have provided the heatmap (Extended Data Fig. 15) of 25 differential expression genes between females and males with Yki and Shd genes labeled.

L498: Please specify the three dated fossil records used for tree calibration.

Response: We apologize for missing this description. We have added the information of the three dated fossil records (*Baissatermes lapideus* (145–99.6 Ma), *Raphogla rubra* (260–251 Ma), and *Rhyniella praecursor* (412.3–391.9 Ma)) in the Method section of the revised manuscript (lines 507-508).

L656: Please provide SRA accession numbers for the raw reads and the genome assemblies.

Response: Thanks for your kind suggestion. We have provided the data accession number (PRJCA016496) for the sequencing data and genome assemblies (lines 675-677).

Fig. 1: I suggest adding ecological photos of these praying mantids to this figure, which will help the readers visualize and understand their adaptive camouflage.

Response: We highly appreciated this important suggestion. We have added the ecological photos of both praying mantises in Fig 1. The four femoral

lobes in combination with the broad abdomen in juvenile orchid mantis give the appearance of five petals of a generalized flower, and dead leaf mantis possesses a broad prothorax with leaf vein on the adult's tucked wings and leaf petiole residue-like legs to mimic a ripped and crumpled leaf (**Response Figure 1**).

Response Figure 1. Photos of the juvenile orchid mantis (left) and adult dead leaf mantis (right).

Extended Data Figures: I noticed that the abbreviations of species names in some figure legends are italicized as well as the species names, but they are not italicized in the figures, so it is recommended to keep the figure legends and figures consistent.

Response: Thanks for your correction. We have canceled the italics of the abbreviations in figure legends to keep them consistent with figures.

Extended Data Fig. 10: The internal nodes of the phylogenetic tree have different colors. Please specify their meanings.

Response: Thanks for your kind suggestion. The colors of internal nodes indicate eight subfamilies of the ATP-binding cassette (ABC) gene family. We have added the description in figure legend.

Reviewer #2 (Remarks to the Author):

Praying mantises (referred to as insect order Mantodea) represent a predatory polyneopteran insect lineage for which a reference genome has not been available so far. A de novo assembly of two mantid genomes has now been provided in the study submitted by Huang et al. Based on two Southeast Asian taxa, the orchid mantis *Hymenopus coronatus* (Hymenopodinae) and the dead leaf imitator *Deroplatys lobata* (Deroplatyidae, not Mantidae as stated by the authors). Both species exhibit an unusual, specialised morphology among praying mantises and indeed can be considered highly charismatic taxa. Based on the genomic data assembled the authors infer the potential developmental genetic base of leg enlargements and colouration that largely contribute to the animals' stunning plant mimicry. Technically there are no obvious flaws or unjustified conclusions (but I am no expert in developmental biology or bioinformatics), but the authors tend to overstate their findings. The de novo assembly of genomes for understudied organismic groups undoubtedly is a worthwhile endeavour per se, however these kind of studies are frequently published in genome report journals and no longer justify publication in high-ranking journals unless the conclusions are of major significance for our understanding of biological phenomena in general. Here, one conclusion (based on transcriptomes/gene expressions of various instars) is that the cuticle is responsible for the leaf-like expansions of the legs and that Wnt signaling determined by Ubx is involved in this process, two observations of interest but not unexpected. To satisfactorily corroborate this, it would have been favourable to compare the results with one of the numerous mantid species that does not exhibit leaf-like leg expansions (as a control!). The two species examined in fact are rather closely related and highly subordinate within the mantid tree of life. Investigation of less related taxa stemming from different major mantid lineages and different geographic regions would also have allowed for more general statements in regard of genomic properties, e.g. gene family expansions/contractions; Trypsin gene family expansion in mantises. These statements cannot be made for all mantises but only for the rather derived narrow lineage that is represented by *Hymenopus* and *Deroplatys*. This is not only a deficiency for statements regarding mantids but for insects in general (Figure 1) when only a single cockroach, a single termite, a single dragonfly etc. is included (it is also difficult to see which insect orders are represented therein). Furthermore, the statement that "the lack of a whole-genome sequence for any Mantodea species has impeded the resolution of the phylogenetic position of this group within Insecta" is simply not true: The 324 single-copy orthologs used for reconstructing the phylogeny of 18 insect species in the present study pales in comparison to the 1478 single-copy nuclear genes used by Misof et al. (2014) for their phylogenomic study of over 100 insect species including mantises and allies. This study is cited and supported by the present data, so it is

unclear what prompted the authors to make this statement. There are further conclusions that just overemphasise the significance of the study when the authors suggest based on the limited temporal demographic estimations “that proper conservation plans should be developed for mantises in response to ongoing climate change” (This statement might just be true for most organisms on this planet anyway).

This said, there is valuable and noteworthy information provided here that should be published, but presented less fulsome and in a more cautious manner. In my opinion, this study is better suited for a a less high-ranking and more specialised journal focussed on genome reports.

Response: Thank you for your insightful comments and the time taken to review our manuscript. We have carefully made revisions to the manuscript according to your comments to make sure all statements were presented in a more cautious manner. We have changed Mantidae into Deroplatyidae, and deleted the statement “that proper conservation plans should be developed for mantises in response to ongoing climate change”.

Camouflage is a classic example of a trait responding to natural selection that has been of broad interest to the evolutionary and developmental biologists. The orchid mantis displays one of the most spectacular cases of camouflage in nature. However, the lack of genomic resources for any Mantodea lineages has impeded further in-depth molecular and developmental investigation. We agree with your comment that a simple genome report is not suitable for publishing in this high-ranking journal, however, this study is not merely a genome report, but a work of evolutionary and developmental study of this highly camouflaged species. We hope more genomic assemblies of different mantis lineages from different geographic regions available in future that will allow to explore more biological phenomena in general.

Comparative genomics has been considered as a powerful tool for studying how the traits of organisms have changed over time by comparing the complete genome sequences of the representative species from different taxa. This study focuses on the orchid mantis *H. coronatus* with the closely related species *D. lobata* as a contrast for comparative genomic analysis to reveal the unique camouflaged characters of the orchid mantis. For the analysis of *Trypsin* gene family expansions/contractions, due to insectivory is a generalizable feature to any mantis, we selected these two praying mantises and the publicly available high-quality genomic assemblies of other different insect lineages with omnivory and herbivory such as *Z. nevadensis*, *B. germanica*, and *D. melanogaster* to define the unique evolutionary characters in mantis. We found significant expansion of *Trypsin* both in *H. coronatus* and *D. lobate* with higher enzyme activity, suggesting the important role of *Trypsin* in insectivory adaptation of mantises. We agree with your comment that a single species can't completely represents the whole taxa, however, the high-quality genome assemblies generally lack so far.

More mantis genome resources available in future will allow to explore the evolutionary character of *Trypsin* in mantis lineage.

To address the phylogenetic position of the orchid mantis, we selected 16 representative species from 15 orders that possess high-quality genomic assemblies for comparative genomic analysis. The classification information of these 16 species has been provided in the revised manuscript (lines 483-493). More divergent taxa usually result in fewer single-copy orthologues. Almudi et al (2020) obtained 1560 single-copy orthologues from 7 insect orders, Ylla et al (2021) obtained 732 single-copy orthologues from 9 insect orders, and Zhang et al (2019) obtained 88 single-copy orthologues from 10 insect orders. Misof et al (2014) obtained 1478 single-copy orthologs from the transcriptome data of 12 species across 9 orders, and the orthologous sequences of remaining 91 species were obtained by mapping the transcripts to 1478 single-copy orthologs. We used the strategy via whole-genome alignment and reciprocal best hit method to identify the genes that were strict 1:1 orthologous among all the 16 orders. Finally, we obtained 324 reliable single-copy orthologs to construct the maximum-likelihood trees, which would yield a robust topology.

To address the genetic basis of “petal-like” femoral expansion, the first-instar larva (L1) was used as the control for comparison with other developmental stages, as the “petal-like” cuticular enlargement is absent in L1 and present from the L2 stage. We found a key transcription factor *Arm* that drives the ventral flat expansion by enhancing cuticle expression, which has never been reported in the leg morphogenesis of hemimetabolus insects before. To better visualize and understand their adaptive camouflage, we have also provided the photos that shows the details of color differences as well as femoral modifications between L1 and other forms in the revised Figure 3A.

Taken together, this study not only provides the first two reference genomic resources of the praying mantis, but also explores the genetic mechanisms of multiple adaptive traits in the orchid mantis via the combination of genomic and transcriptomic analysis with functional testing, which has far-reaching implications for future in-depth study of Mantodea. We believe this study will contribute to understanding camouflage phenomena of the praying mantis.

Reference:

- Almudi I, et al. *Genomic adaptations to aquatic and aerial life in mayflies and the origin of insect wings. Nat Commun. 2020 May 26;11(1):2631.*
- Zhang X, et al. *Penaeid shrimp genome provides insights into benthic adaptation and frequent molting. Nat Commun. 2019 Jan 21;10(1):356.*
- Ylla G, et al. *Insights into the genomic evolution of insects from cricket genomes. Commun Biol. 2021 Jun 14;4(1):733.*

Reviewer #3 (Remarks to the Author):

In this study, Huang et al. focus on unique aspects of camouflage morphology in orchid mantids, mainly aposematic coloration at L1 stage and “petal-like” femoral enlargement in adults. To begin to understand mechanisms responsible for these phenotypic innovations, the authors first generate de novo assembly of two chromosome-level genomes (one for the orchid mantis, and one for its close relative, the dead leaf mantis). Then, by using comparative genomic analysis authors identify Scarlet gene as a key player for mantis camouflage coloration. Finally, functional testing via RNAi, reveals that the “petal-like” morphology is regulated by WNT signaling. Overall, this is excellent and very substantial study, which provides novel insights into the origins of morphological novelties in an exciting new model insect (orchid mantis). The quality and the amount of work is exceptional and text is easy to read and follow. One issue that has to be corrected, though, is authors’ simplistic interpretation of the role of *Ubx* at L1 stage and its putative role in inhibiting Wnt signalling (for details see below). Aside from that, this is indeed a very nice work that will be of broad interest to a large audience of evolutionary and developmental biologists.

Aleksandar Popadić

Response: Thank you for your constructive comments and the detailed advice on the role of *Ubx* that is very important for improving our manuscript. The additional experiment was conducted accordingly to address the role of *Ubx* and the corresponding statement has been corrected. Below is our point-to-point response to each of the comments.

Main concern:

1) The authors present interpretation regarding the role of *Ubx* at L1 stage is likely incorrect, or at least greatly speculative. This is due to their reliance on several *Drosophila* papers (Diaz-de-la-Loza et al. 2020; Oberhofer et al. 2014), which are not relevant for hemimetabolous insects such as mantids. For that matter, the cited *Tribolium* paper (Lewis et al. 2000) is also inappropriate as it deals specifically with the development of A1 appendage, and not the T3 leg. Instead, the authors should get acquainted with now classic work that shows that *Ubx* regulates T3 leg enlargement in hemimetabolous insects (Mahfooz et al 2004; Mahfooz et al 2007). Alternatively, it can also enlarge T2 leg (Khila et al. 2009).

If the orchid mantis follows the general trend observed in other mantids, then its T3 tibia and tarsus are enlarged compared to their T2 counterparts (at L1 stage; see Mahfooz et al 2004). If observations in *Tenodera* apply to *Hymenopus* and *Deroplatys* then *Ubx* is expressed specifically in T3 tibia and tarsus, and not in the femur in the latter two species. In other words, the

authors' entire explanation about the role of Ubx becomes false, as Ubx is not expressed in T3 femur. Consequently, the experimental evidence highlighted in the paper is misleading as it refers to Ubx mRNA level sampled from the entire leg (including tibia and tarsus).

The authors should do the following: repeat the experiment and use individual leg segments as a tissue source (T3 femur; T3 tibia; and T3 tarsus).

Alternatively, they can perform antibody staining on Hymenopus embryos using a cross-reacting UbdA antibody (Kelsh et al 1994). Either would prove that Ubx is not localized in T3 femur, and hence, is not responsible for the absence of petal-like morphology at L1 stage.

Note: the femoral lateral extensions are not restricted to T3 leg only, but are present on all three leg pairs – this is another evidence that Ubx is not involved in these modifications as it is never expressed in T1 leg (and only very rarely in T2 leg).

Response: We are very grateful for your detailed comments. We have conducted additional experiments following your suggestion. We have compared the size of T2 and T3, and detected the expression level of *Ubx* in femur, tibia and tarsus, respectively. Consistent with the trend observed in *Tenodera aridifolia* (Mahfooz et al 2004), we found that the T3 tibia and tarsus of *H. coronatus* were enlarged both in length and area with higher expression level of Ubx mRNA compared to their T2 counterparts (**Response Figure 2**). In term of the development of the hind leg, the expression pattern of *Ubx* in *H. coronatus* follows the general trend observed in hemimetabolous insects (Mahfooz et al 2004; Mahfooz et al 2007).

Response Figure 2. Comparison in length (A) and area (B) of the femur, tibia and tarsus of T2 and T3 legs at L1 and L2 stages, and detection of the expression level of the two transcripts of Ubx at L1 stage (C). Ten replicates for A and B, and three replicates for C. * $p < 0.05$ and ** $p < 0.01$ by *t*-test.

Contrast to *T. aridifolia*, we detected a higher expression level of Ubx in T3 femur than that in T2, consistent with the observation of more enlarged T3 femur of *H. coronatus* (Response Figure 2). Compared to *T. aridifolia*, *H. coronatus* exhibits “petal-like” femoral expansion from onset of L2 stage. We also found that a high mRNA expression of Ubx in femur lasted until L2 stage in T2 and T3, which might contribute to the occurrence of “petal-like” cuticular enlargement (Response Figure 3). These additional data suggest that the prolonged expression of Ubx in femur was associated with the petal-like enlargement.

Response Figure 3. Comparison of the *Ubx* mRNA level in the femur of T2 and T3 legs between L1 and L2 stages. * $p < 0.05$ and ** $p < 0.01$ by *t*-test.

Thus, the prolonged expression of *Ubx* in femur in the early developmental stage plays a key role in the modulation of the petal-like enlargement. We have corrected the statement on the role of *Ubx* in the revised manuscript (lines 261-275, revised Fig. 3 and Extended Data Fig. 16). The corresponding references were also updated.

2) In the light of the importance of the “petal-like” cuticular enlargement, a more detailed visual presentation should be provided. Authors should create a new Fig 1 that shows both the details of color differences between L1 and adult forms, as well as femoral modifications from L2-adult stage. This figure should encompass images of L1, L2, and adult stages. In addition, it should display dissected T2 and T3 legs in L1 (showing absence of “petal-like” morphology), then dissected T2 and T3 legs in L2 and adult stages (showing the presence of this trait).

Response: We highly appreciate this very important suggestion. The photos of T2 and T3 legs of L1, L2, L5 and adult individuals were added and integrated into Figure 3A as this part interprets femoral morphogenesis (**Revised Fig. 3A**).

Revised Fig. 3A Photos of T2 and T3 legs showing the differences in color and shape between L1 and the later developmental stages. Scale bar = 1 mm.

REVIEWERS' COMMENTS

Reviewer #1 (Remarks to the Author):

My comments have been properly addressed by the author. I am satisfied with the current version of this manuscript and I consider it acceptable for publication.

Reviewer #2 (Remarks to the Author):

The authors have addressed all points raised in my previous review, and improved the manuscript in numerous aspects. However, the issues regarding the phylogeny of Mantodea are not explained/revised to my satisfaction: There are more substantial trees available who have already resolved the phylogenetic position of Mantodea among Polyneoptera, so the (unaltered) following statement is not justified: "The lack of a whole-genome sequence for any Mantodea species has impeded the resolution of the phylogenetic position of this group within Insecta. To determine the phylogenetic position of Mantodea, 324 single-copy orthologs identified in these two mantises and 16 other insects from 15 key extant insect orders were used to construct a phylogenetic tree (Fig. 1E, Extended Data Fig. 6). The phylogenetic result was congruent with a recent insect phylogeny" The phylogeny of Mantodea/its position among insects is clarified and has just been corroborated here. So how can the lack of the data provided here by the authors have impeded "resolution of the phylogenetic position"? This is a false statement that is also contradictory in the cited paragraph itself since the presented data do corroborate a massive phylogeny that has been published here ago. Furthermore, Fig. 1e still has no information on the orders of the insects on the tree, just small pics that are nice to have but difficult to apprehend by the non-expert (the dragonfly is easy, but even the stick insect is difficult to recognise - by the way misspelled: it must be *Clitarchus hookeri*).

Reviewer #3 (Remarks to the Author):

In their revised manuscript, the authors have completely addressed all of the original comments regarding the early role of Ubx in legs developments (mainly T3 vs T2). Congratulations on a very nice study.

Aleksandar Popadić

Reviewer #1 (Remarks to the Author):

My comments have been properly addressed by the author. I am satisfied with the current version of this manuscript and I consider it acceptable for publication.

Response: Thanks for your constructive comments for improving our manuscript.

Reviewer #2 (Remarks to the Author):

The authors have addressed all points raised in my previous review, and improved the manuscript in numerous aspects. However, the issues regarding the phylogeny of Mantodea are not explained/revised to my satisfaction: There are more substantial trees available who have already resolved the phylogenetic position of Mantodea among Polyneoptera, so the (unaltered) following statement is not justified: "The lack of a whole-genome sequence for any Mantodea species has impeded the resolution of the phylogenetic position of this group within Insecta. To determine the phylogenetic position of Mantodea, 324 single-copy orthologs identified in these two mantises and 16 other insects from 15 key extant insect orders were used to construct a phylogenetic tree (Fig. 1E, Extended Data Fig. 6). The phylogenetic result was congruent with a recent insect phylogeny" The phylogeny of Mantodea/its position among insects is clarified and has just been corroborated here. So how can the lack of the data provided here by the authors have impeded "resolution of the phylogenetic position"? This is a false statement that is also contradictory in the cited paragraph itself since the presented data do corroborate a massive phylogeny that has been published here ago. Furthermore, Fig. 1e still has no information on the orders of the insects on the tree, just small pics that are nice to have but difficult to apprehend by the non-expert (the dragonfly is easy, but even the stick insect is difficult to recognise - by the way misspelled: it must be *Clitarchus hookeri*).

Response: Thanks for your constructive comments. We have changed the statement in the revised manuscript to "Phylogenomic analysis was performed to test the phylogenetic position of Mantodea". We have revised the spell of *C. hookeri* in the figure 1e, and made sure that the spell in the main text is correct. We have removed the insect photos in figure 1e to add the order information of each insect.

Reviewer #3 (Remarks to the Author):

In their revised manuscript, the authors have completely addressed all of the original comments regarding the early role of Ubx in legs developments (mainly T3 vs T2). Congratulations on a very nice study.

Aleksandar Popadić

Response: Thanks for your constructive comments for improving our manuscript.